# Equilibrium, Kinetic, and Thermodynamic Studies of Cationic Dyes Adsorption on Corn Stalks Modified by Citric Acid

**Liudmyla Soldatkina \*** and **Marianna Yanar**

Faculty of Chemistry and Pharmacy, Odesa I.I. Mechnikov National University, 2, Dvoryanska Str., 65082 Odesa, Ukraine; mariannayanar@onu.edu.ua
\* Correspondence: soldatkina@onu.edu.ua; Tel.: +38-048-723-82-64

**Abstract:** The modification of agricultural wastes and their use as low-cost and efficient adsorbents is a prospective pathway that helps diminish waste and decrease environmental problems. In the present research, the natural adsorption capacity of corn stalks (CS) was improved by modification of their surface with citric acid. The adsorption capacity of the modified corn stalks (CS-C) was determined with the help of cationic dyes (methylene blue and malachite green). The equilibrium, kinetics, and thermodynamics of the cationic dyes on CS-C were studied. The Langmuir isotherm model best fitted the data both for methylene blue and malachite green adsorption on CS-C. The adsorption kinetics of the cationic dyes was well described by the pseudo-second order model. Thermodynamic studies revealed that adsorption of the cationic dyes on CS-C was an endothermic process. Negative results of $\Delta G^o$ (between $-31.8$ and $-26.3$ kJ mol$^{-1}$) indicated that the adsorption process was spontaneous in all the tested temperatures. The present study verified that citric acid-modified corn stalks can be used as a low-cost and effective adsorbent for removal of cationic dyes from aqueous solutions.

**Keywords:** modified corn stalks; citric acid; cationic dyes; adsorption; equilibrium; kinetics; thermodynamics

## 1. Introduction

Various industries, such as textile, pulp and paper, tannery, pharmaceutical, food processing, cosmetics, etc., employ synthetic dyes to color their products [1,2], but plants and factories of these industries release a considerable amount of colored wastewater into the environment. The presence of a very small amount of dye (<1.0 mg/L) can markedly color the water and have a serious impact on its quality and the transparency of water bodies, rendering them unfit for the aquatic environment [3]. According to [4], industrial textile effluents of dyeing stages consist of 10–50 mg L$^{-1}$ of cationic dyes. The removal of dyes from wastewater is important in protecting the environment because they usually have a complex aromatic structure that makes them stable to light, heat, and oxidizing agents while being recalcitrant to biodegradation [3]. The synthetic dyes also pose serious threats to human health since they are toxic, teratogenic, mutagenic, and carcinogenic and cause severe damage to the central nervous system, digestive system, and liver of human bodies, even at low concentrations [5].

At the present time, different technologies (coagulation, electrocoagulation, flocculation, adsorption, ion exchange, membrane processes, irradiation and oxidative processes, biological treatment) have been developed for dye removal with varying levels of success [1,2,5]. Unfortunately, most of the known technologies are generally ineffective in dye removal, expensive or time-consuming, and less adaptable to a wide range of dye wastewater. Amongst these technologies of dye removal, adsorption is believed to be the most efficient method because of its convenience, ease of operation, flexibility, and simplicity of design [3,6]. Activated carbons, which are used to remove organic compounds

from wastewater due to their excellent adsorption ability, are the most explored adsorbents. Despite the popularity of activated carbons as adsorbents, their use is limited due to the expensive production and regeneration. Therefore, the adsorption treatment will become less expensive if the effective adsorbents are made from cheap and available materials.

Currently, in many countries, lignocellulosic materials as the wastes of large-scale industrial processes and agricultural waste materials are considered potential precursors for the preparation of low-cost and effective adsorbents [7], because they are generated on a large scale, available, and inexpensive.

Generally, native lignocellulosic materials are rarely directly used for adsorption due to their low adsorption capacity, which can be increased using the modification of different chemicals and chemical reactions (acid, alkali, surfactants, cold plasma, etherification, esterification, and the graft copolymerization process [1]). However, it is expedient to obtain adsorbents from lignocellulosic materials using chemicals that satisfy the following requirements: efficient, economical, environmentally friendly, and easily produced [8].

Various citric acid-modified lignocellulosic materials used for the removal of cationic dyes are given in Table 1 [3,6,8–22]. The modification of lignocellulosic materials allowed the content of carboxyl groups to be increased through an esterification reaction, and they were used as adsorbents for removal of the cationic dyes. For example, Fathy et al. found that the maximum adsorption capacities of unmodified and modified rice straw are 32.6 and 135.1 mg g$^{-1}$ [11]. Thus, the citric acid modification of rice straw allowed its adsorption capacity to increase by ~4 times. Moreover, citric acid is an inexpensive, nontoxic, biocompatible, water-soluble, and mild organic acid.

**Table 1.** Adsorption capacities of citric acid-modified lignocellulosic materials towards cationic dyes.

| Adsorbent | Dye | Maximum Adsorption Capacity [1] (mg g$^{-1}$) | Best Kinetic Model | Best Isotherm Model | Nature | Reference |
|---|---|---|---|---|---|---|
| Rice straw | Malachite Green | 256.41 (293 K) | Pseudo-First order | Langmuir and Freundlich | - | [9] |
| Rice straw | Methylene Blue | 270.3 (293 K) | Pseudo-first order | Langmuir | - | [10] |
| Rice straw | Methylene Blue | 135.1 (308 K) | Pseudo-second order | Tempkin | Spontaneous | [11] |
| Rice straw | Methylene Blue | 164.16 (298 K) | Pseudo-second order | - | - | [12] |
| Wheat straw | Methylene Blue | 312.5 [2] | Pseudo-second order | Langmuir | Endothermic and Spontaneous | [13] |
| Wheat straw | Methylene Blue | 396.9 (293 K) 432.8 (303 K) 450.0 (323 K) | Pseudo-second order | Langmuir and Freundlich | Endothermic and Spontaneous | [14] |
| Wheat straw | Crystal Violet | 227.27 [2] | Pseudo-second order | Langmuir | Endothermic and Spontaneous | [13] |
| Wheat bran | Malachite Green | 67.547 (293 K) 64.781 (303 K) 61.165 (323 K) | Pseudo-second order | Langmuir | Exothermic and Spontaneous | [6] |
| Sesame straw | Methylene Blue | 650 (298 K) | Pseudo-second order | Langmuir | Endothermic and Spontaneous | [8] |
| Kenaf core fibers | Methylene Blue | 103.1 (293 K) 128.2 (313 R) 131.6 (333 K) | Pseudo-second order | Langmuir | Endothermic and Spontaneous | [15] |
| Leaves of Ricinus communis | Methylene Blue | 333.33 [2] | Pseudo-second order | Temkin | - | [16] |

**Table 1.** *Cont.*

| Adsorbent | Dye | Maximum Adsorption Capacity [1] (mg g$^{-1}$) | Best Kinetic Model | Best Isotherm Model | Nature | Reference |
|---|---|---|---|---|---|---|
| Peanut shell | Methylene Blue | 120.48 (303 K) 119.05 (313 K) 108.69 (323 K) | Pseudo-second order | Freundlich | Exothermic and Spontaneous | [17] |
| Peanut shell | Methylene Blue | 99.41 (283 K) 111.38 (298 K) 129.83 (313 K) | Pseudo-second order | Langmuir | Endothermic and Spontaneous | [18] |
| Peanut shell | Neutral Red | 112.72 (283 K) 131.8 6(298 K) 151.24 (313 K) | Pseudo-first order | Freundlich | Endothermic and Spontaneous | [18] |
| Seeds of Abelmoschus esculentus | Gentian Violet | 211.46 (288 K) 245.82 (303 K) 253.29 (318 K) | Pseudo-first order | Redlich-Peterson | Endothermic and Spontaneous | [3] |
| Grass | Methylene Blue | 301.1 [2] | Pseudo-first order | Langmuir | - | [19] |
| Peach stone | Methylene Blue | 178.25 (303 K) | Pseudo-second order | Freundlich | - | [20] |
| Barley straw | Methylene Blue | - | Pseudo-second order | - | - | [21] |
| Jerusalem artichoke stalks | Methylene Blue | - | Pseudo-second order | - | - | [21] |
| Corn stalks | Malachite Green | 16.47 [2] | Pseudo-second order | Langmuir | - | [22] |

[1] Maximum adsorption capacities were obtained from Langmuir model. [2] Unknown temperature.

Among various agro-industrial waste, corn stalks are generated in enormous amounts in the production of corn, which is quite extensively planted in many countries (USA, China, Brazil, Argentina, Ukraine, India, etc.). Corn stalks are traditionally used for animal bedding or directly incinerated [5]. Corn stalks consist of approximately 40% of cellulose; 18% of lignin; 12–26% of pentosans; 3.5% of tar, waxes, and fats extracted with an alcohol-benzene mixture; 12% of substances extracted with hot water; and 25% of substances extracted with a 1% alkaline solution [22–24]. Furthermore, corn stalks have a macroporous network structure, good mechanical stability, and various functional groups (hydroxyl, carboxyl, carbonyl).

In this paper the equilibrium, kinetic, and thermodynamic studies of adsorption of cationic dyes onto citric acid-modified corn stalks from an aqueous solution were investigated because this information is currently quite limited. Methylene blue and malachite green were studied as cationic dyes due to their wide application and difference in chemical structure.

## 2. Materials and Methods

### 2.1. Materials

Chemical reagents as $C_6H_8O_7 \cdot H_2O$, NaOH, HCl, and $CH_3COOH$ were obtained from Cherkassy State Chemical Plant (Chercassy, Ukraine) and had a pure analytical quality. Standard buffer solutions (pH = 1.68; 6.86; 9.18) were obtained from Kyiv Plant of Reagents, Indicators and Analytical Products "RIAP" (Kyiv, Ukraine). Cationic dyes (methylene blue (MB) and malachite green (MG)) were purchased from Ukrainian company "Fine organic synthesis plant "Barva AG"". Methylene blue is a dye of thiazine type ($C_{16}H_{18}N_3SCl$, C. I. No. 52015, MW = 319.5 g mol$^{-1}$, $\lambda_{max}$ = 665 nm). Malachite green (MG) is a dye of diaminotriphenylmethane type ($C_{23}H_{25}N_2Cl$, C.I. No 42000, MW = 364.5 g mol$^{-1}$, $\lambda_{max}$ = 617 nm).

Methylene blue and malachite green have been continually investigated by researchers in adsorption studies of cationic dyes from aqueous solutions due to their wide application. Methylene blue is used for the paper coloring, cotton and wool dyeing, paper stock coating, calico printing, cotton dyeing, and leather printing and dyeing [16]. Malachite green has applications including dyeing of cotton, jute, silk, wool, and leather and is extensively employed all over the world in the fish farming industry as fungicide, ectoparasiticide, and disinfectant [9].

The stock solutions of methylene blue and malachite green (1000 mg L$^{-1}$) were prepared in distilled water. All experimental solutions were prepared by diluting the stock solutions of the cationic dyes with distilled water to the required concentration

Corn stalks were obtained from the countryside in the Odesa region, Ukraine; dried at room temperature; and then ground in the electrical universal grinder and screened to obtain the fraction <250 μm.

### 2.2. Methods

#### 2.2.1. Preparation of Adsorbent

The samples of unmodified and modified corn stalks from the reference [25] were used in this study. The citric acid-modified corn stalks were prepared when full factorial central composite design (as $2^3$) had been employed for the optimization of corn stalk modification.

Prior to the adsorption studies, ground corn stalks were soaked in 0.1 M NaOH at the ratio of 1:20 (stalks/alkali, *w/v*) at 293 K for 120 min. Corn stalks were separated by filtration and rinsed with distilled water to remove residual alkali. Then, corn stalks were dried in the oven at 323 K up to constant weight and preserved in a desiccator for further modification by citric acid. The prepared corn stalks were mixed with 0.78 M citric acid at the ratio of 1:20 (stalks/acid, *w/v*) and shaken at 150 rpm at 293 K for 1 h. Then, the chemical reaction between citric acid and corn stalks was proceeded by raising the oven temperature to 393 K for 210 min. After cooling, the modified corn stalks were washed with distilled water, filtered, and dried at 323 K to constant weight. The obtained adsorbent (CS-C) was preserved in a desiccator for further use.

The characteristics of CS and CS-C are presented in Table 2.

**Table 2.** Characteristics of CS and CS-C [25].

| Adsorbent | Specific Surface Area (m$^2$ g$^{-1}$) | pH$_{pzc}$ | COOH (mmol g$^{-1}$) |
|:---:|:---:|:---:|:---:|
| CS | 22.1 | 5.4 | 0.7 |
| CS-C | 45.3 | 3.3 | 3.5 |

#### 2.2.2. Infrared Analysis

The functional groups present in the native and modified corn stalks were characterized by a Fourier transform infrared (FT-IR) spectrophotometer (Perkin-Elmer Spectrometer, Waltham, MA, USA) with a resolution of 0.4 cm$^{-1}$. In each case, 1.0 mg of dried sample and 100 mg of KBr were homogenized using mortar and pestle and thereafter pressed into a pellet. Eight scans were the reported spectra averaged for 1 min.

#### 2.2.3. Adsorption Studies

The batch adsorption experiments were carried out under atmospheric conditions in a rotary shaker (Elpan type 357, Lubawa, Poland) at 150 rpm and a constant temperature (293, 313, and 333 K). Adsorption equilibrium studies were conducted by contacting 10 cm$^3$ of dye solutions with different initial concentrations (20–1000 mg L$^{-1}$) and 0.1 g of CS-C in 50 cm$^3$ conical flasks for 90 min (equilibrium time was reached). Kinetic experiments were carried out by varying contact time from 5 to 210 min at 35 mg L$^{-1}$ of the dye concentration and 5 g of CS-C in 500 cm$^3$ of the dye solution.

The pH adjustments of the solutions were made by addition of HCl (0.1 M) or NaOH (0.1 M) using a pH meter (Universal ionomer EV−74, Gomel, Belarus) calibrated with standard buffer solutions.

Our preliminary results showed that adsorption of cationic dyes on CS-C had maximum values at pH = 6.0–10.0, while at lower pH values, the adsorption of cationic dyes on CS-C was decreasing (the figure is not shown). It was also taken into account that according to [4], the cationic dyes removal is only by adsorption at pH = 6, and in an alkaline environment (pH range between 8.0 and 11.0), the color change of the dye solution occurs due to a chemical reaction between the cationic dye and OH- ions. Therefore, all adsorption experiments for dye removal using CS-C adsorbent were carried out at a pH value of 6.0. This pH value is in line with the studies reported on the adsorption removal of gentian violet on seeds of Abelmoschus esculentus modified by citric acid [3] and methylene blue on peanut shell modified by citric acid [18] and rice straws modified by citric acid [12].

After adsorption, the adsorbent was separated using centrifugation at 4000 rpm for 10 min. The amount of the dye on the adsorbent ($q$, mg/g), which represents the dye uptake, was calculated from the difference in dye concentration in the aqueous phase before and after adsorption, according to Equation (1):

$$q = \frac{C_o - C}{m} \cdot V \tag{1}$$

where $Co$ is the initial dye concentration, mg L$^{-1}$; $C$ is the dye concentration after adsorption at time $t$, mg L$^{-1}$; $m$ is the mass of adsorbent, g; and $V$ is the volume of the dye solution, L.

The concentrations of methylene blue and malachite green in liquid phase before and after adsorption were estimated by monitoring the change in absorbance values at maximum wavelengths of 665 and 617 nm, respectively, using a UV/VIS spectrophotometer (SF-56, Spectral, LOMO, St. Peterburg, Russia). The amount of adsorbed dyes was obtained using the calibration curves in the range of 1–10 mg L$^{-1}$. If after adsorption the absorbance of dye solution was too high, we used a dilution of solution that had an absorbance in range of the calibration plot.

When time $t$ is equal to the equilibrium time ($t_e$), the amount of the dye on the adsorbent at equilibrium time ($q_e$) was calculated using Equation (1).

In order to ensure the reproducibility of the results, all the adsorption experiments were performed in triplicate, and the average values were used in data analysis. Relative standard deviations were found to be within ±3%.

### 2.2.4. Desorption Studies

The adsorbent CS-C (1 g) was added into 100 cm$^3$ of the dye solution (200 mg L$^{-1}$) at pH = 6 and 293 K with stirring at 150 rpm for 120 min. Dye-loaded adsorbent was washed with distilled water to remove any unabsorbed dye and dried at 323 K. Desorption of the dyes from CS-C was carried out using one cycle by immersing 0.1 g of the dye-loaded adsorbent into 10 cm$^3$ distilled H$_2$O, 0.1 M HCl, 0.2 M HCl, 0.1 M NaOH, and 0.1 M CH$_3$COOH, respectively, at 293 K with stirring at 150 rpm for 120 min. The efficiency of dye desorption removal was calculated by Equation (2):

$$S = \frac{C_d \cdot V_d}{q_e \cdot m} \cdot 100\% \tag{2}$$

where $q_e$ is the equilibrium amount of the dye adsorbed on the adsorbent, mg g$^{-1}$; $C_d$ is the dye concentration in solution after desorption, mg L$^{-1}$; $V_d$ is the volume of the eluent, L; and $m$ is the mass of the adsorbent, g.

### 2.2.5. Kinetic Studies

The study of adsorption kinetics arouses considerable interest in predicting the controlling of the adsorption process. In this paper, the kinetic parameters of the four mod-

els, viz., pseudo-first order (Equation (3)), pseudo-second order) Equation (4)), Elovich (Equation (5)), and Weber–Morris (intraparticle diffusion) (Equation (6)), were used to obtain the best kinetic curves:

$$ln(q_e - q) = lnq_e - k_1t \tag{3}$$

$$\frac{t}{q} = \frac{1}{k_2q_e^2} + \frac{t}{q_e} \tag{4}$$

$$q = \frac{1}{\beta}ln(\alpha\beta) + \frac{1}{\beta}lnt \tag{5}$$

$$q = k_{id}t^{1/2} + I \tag{6}$$

where $q_e$ and $q$ are the amount of the dye on the adsorbent at equilibrium and various time $t$, respectively, mg g$^{-1}$; $k_1$ is the adsorption rate constant of pseudo-first order model, min$^{-1}$; $k_2$ is the adsorption rate constant of pseudo-second order model, g mg$^{-1}$ min$^{-1}$; $\alpha$ is the initial adsorption rate, mg g$^{-1}$ min$^{-1}$; $\beta$ is the desorption constant, g mg$^{-1}$; $k_{id}$ is the adsorption rate constant of the intraparticle diffusion model, mg g$^{-1}$ min$^{-1/2}$; and $I$ is the constant, which gives an idea about the boundary layer thickness, mg g$^{-1}$.

2.2.6. Adsorption Isotherms

In the present study, two-parameter isotherm models, viz., Freundlich (Equation (7)), Langmuir (Equation (8)), and Temkin (Equation (9)), were used to examine the relationship between the amount of adsorbed dye and its equilibrium concentration:

$$\frac{C_e}{q_e} = \frac{1}{q_mK_L} + \frac{C_e}{q_m} \tag{7}$$

$$lnq_e = lnK_F + \frac{1}{n}lnC_e \tag{8}$$

$$q_e = \frac{RT}{b}lnK_T + \frac{RT}{b}lnC_e \tag{9}$$

where $C_e$ is the equilibrium dye concentration in solution, mg L$^{-1}$; $K_F$ is the Freundlich constant related to the adsorption capacity, mg$^{1-1/n}$ L$^{1/n}$ g$^{-1}$; $1/n$ is the adsorption intensity; $q_m$ is the monolayer capacity of the adsorbent, mg g$^{-1}$; $K_L$ is the Langmuir constant that relates to energy of adsorption, L mg$^{-1}$; $K_T$ is the Temkin equilibrium constant corresponding to the maximum binding energy, L g$^{-1}$; and $b$ is the heat of adsorption, kJ mol$^{-1}$.

2.2.7. Adsorption Thermodynamics

The fundamental thermodynamic parameters of adsorption, such as standard Gibbs free energy change ($\Delta G^o$), standard enthalpy change ($\Delta H^o$), and standard entropy change ($\Delta S^o$), are calculated by Equations (10)–(12):

$$\Delta G^o = -RTlnK^o \tag{10}$$

$$K^o = K_L \cdot \gamma \cdot \frac{s_o}{s} \tag{11}$$

$$lnK^o = \frac{\Delta S^o}{R} - \frac{\Delta H^o}{R} \cdot \frac{1}{T} \tag{12}$$

where $\Delta G^o$ is the standard Gibbs free energy change, J mol$^{-1}$; $R$ is the universal gas constant ($R$ = 8314 J mol$^{-1}$ K$^{-1}$); $T$ is the absolute temperature, K; $K^o$ is the dimensionless thermodynamic adsorption constant; $K_L$ is the Langmuir constant, L mol$^{-1}$; $\gamma$ is the number of moles of pure water per liter, mol L$^{-1}$; $s_o$ and $s$ are the van der Waals areas of solvent and dye molecules; $\Delta S$ is the standard entropy change, J mol$^{-1}$ K$^{-1}$; and $\Delta H^o$ is the standard enthalpy change, J mol$^{-1}$.

- Van der Waals areas of solvent and dye molecules were calculated using software package ChemAxon Marvin 5.2 [26]: $s_o$ ($H_2O$) = 0.0959 nm$^2$, $s$ (MB) = 1.01 nm$^2$, $s$ (MG) = 1.19 nm$^2$.
- The values of $\Delta S^o$ and $\Delta H^o$ were evaluated from the intercept and slope of the van 't Hoff plot of ln $K^o$ vs. $1/T$, respectively [17], assuming that $\Delta H^o$ and $\Delta S^o$ are temperature independent from 293–333 K.

### 2.2.8. Error Analysis

In this study, standard error (*SE*) and chi-square test ($\chi 2$) were employed to determine the best-fit model for the experimental adsorption kinetic and equilibrium data.

These error functions are given as

$$SE = \sqrt{\frac{\sum_{i=1}^{N}\left(q_{i,calc} - q_{i,exp}\right)^2}{N-2}} \tag{13}$$

$$\chi^2 = \sum_{i=1}^{N}\frac{\left(q_{i,calc} - q_{i,exp}\right)^2}{q_{i,calc}} \tag{14}$$

where $q_{i,calc}$ is the theoretical amount of the adsorbed dye on the adsorbent, which was calculated from one of the isotherm or kinetic model equations, mg g$^{-1}$; $q_{i,exp}$ is the experimental amount of the adsorbed dye on the adsorbent, mg g$^{-1}$; and $N$ is the number of the data points.

## 3. Results and Discussion

### 3.1. Characterization of Adsorbent CS-C

The FT-IR technique is a useful tool to identify some characteristic functional groups on the surface of the adsorbent material [14,17]. A comparison of the FT-IR spectra of CS and CS-C is shown in Figure 1.

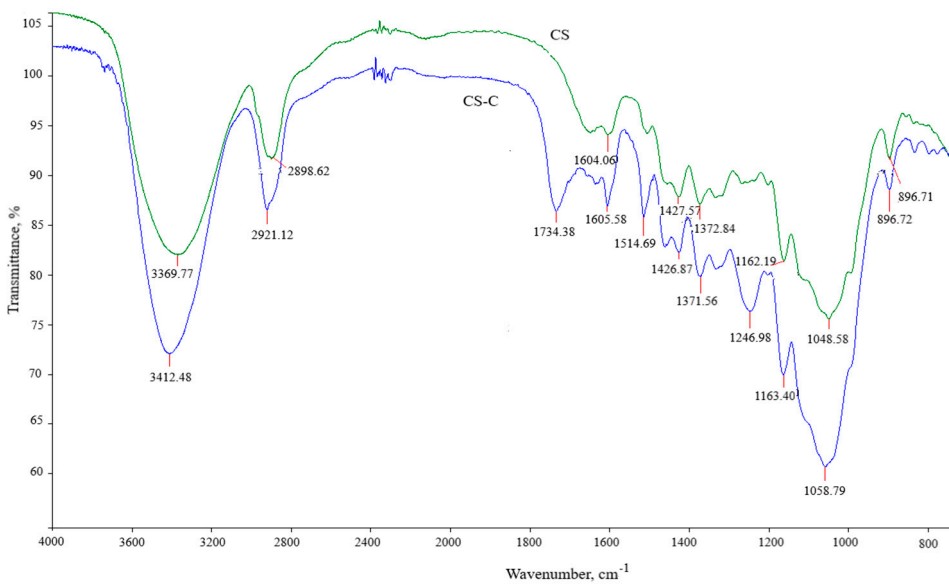

**Figure 1.** FT-IR spectra for unmodified (CS) and citric acid-modified (CS-C) corn stalks.

FT-IR spectra showed (Figure 1) the characteristic broad bends around 3412 and 3370 cm$^{-1}$ in CS and CS-C due to the stretching vibration of hydroxyl groups from macromolecules such as cellulose, hemicellulose, lignin, and adsorbed water [17,27]. The peaks at 2921 and 2899 cm$^{-1}$ in CS and CS-C, respectively, represent the CH stretching and bending vibration of methyl and methylene groups in cellulose and hemicellulose [27]. The peaks

associated with the stretch vibration in aromatic rings were verified at 1604 and 1605 cm$^{-1}$ in CS and CS-C, respectively [14]. Compared with the FT-IR spectra of CS and CS-C, it could be seen that there was a strong characteristic stretching vibration absorption band of the carboxyl group at 1734 cm$^{-1}$ in the FT-IR spectra of CS-C. This reflected the result of citric acid esterification [9,10].

It was well indicated from the FT-IR spectra of CS-C that carboxyl and hydroxyl groups were present in abundance. These groups may function as proton donors; hence, deprotonated hydroxyl and carboxyl groups may be involved in coordination with positive ions of cationic dyes [14].

### 3.2. Kinetic Studies

The adsorption experimental kinetic curves of the cationic dyes on CS-C at 293–333 K were presented in Figure 2. The adsorption process of methylene blue and malachite green on CS-C was rapid at the beginning and then increased slowly until the plateau of adsorption equilibrium was achieved. The fast removal of the dyes at the beginning may be attributed to (i) the rapid attachment of dye ions to the surface of the adsorbent and (ii) the number of binding sites available, consequently leading to an increase in driving force of the concentration gradient between adsorbate in solution and adsorbate–adsorbent interaction [11]. The following slower adsorption process with increasing contact time may be due to the intraparticle diffusion [28].

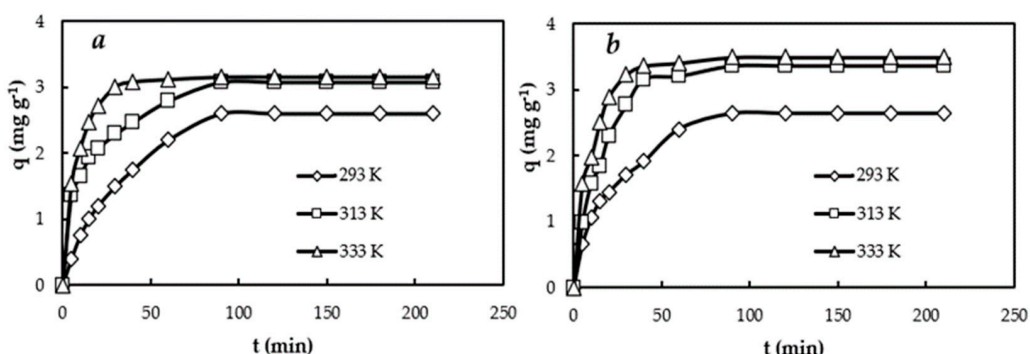

**Figure 2.** Kinetic adsorption curves of MB (**a**) and MG (**b**) on CS-C at different temperatures.

In addition, the adsorption equilibrium of the cationic dyes on CS-C at all temperatures tested was reached in about 90 min, which was faster than the other citric-modified lignocellulosic materials. For example, Gong et al. [10,13] found that the adsorption equilibrium time for the removal of methylene blue on citric acid-modified rice straw was about 600 min, but the adsorption equilibrium time for the removal of methylene blue and crystal violet on citric acid-modified wheat straw was about 300 min.

The adsorption experimental kinetic curves of the cationic dyes on CS-C were analyzed, and calculated parameters from adsorption kinetic models (pseudo-first order, pseudo-second order, Elovich, and Weber–Morris) are summarized in Table 3.

As can be seen from Table 3, the value of $q_e^{exp}$ increased when the temperature increased for both methylene blue and malachite green adsorption. This indicates that a higher temperature favors the adsorption process by increasing the adsorption capacity of CS-C.

<p align="center">**Table 3.** Comparison of the kinetic models.</p>

| Kinetic Model | Parameter | MB | | | MG | | |
|---|---|---|---|---|---|---|---|
| | | 293 (K) | 313 (K) | 333 (K) | 293 (K) | 313 (K) | 333 (K) |
| Experimental data | $t_e$ (min) | 90 | 90 | 90 | 90 | 90 | 90 |
| | $q_e^{exp}$ (mg g$^{-1}$) | 2.60 | 3.07 | 3.15 | 2.63 | 3.36 | 3.49 |
| Pseudo-first order | $q_e^{calc}$ (mg g$^{-1}$) | 2.57 | 1.94 | 1.91 | 2.43 | 3.00 | 2.25 |
| | $k_1 \cdot 10^2$ (min$^{-1}$) | 2.98 | 3.12 | 7.30 | 3.62 | 5.45 | 6.09 |
| | $R^2$ | 0.9918 | 0.9924 | 0.9578 | 0.9628 | 0.9416 | 0.9420 |
| | SE | 0.09 | 1.23 | 1.32 | 0.24 | 0.38 | 1.32 |
| | $\chi^2$ | 0.05 | 15.7 | 11.61 | 0.54 | 0.60 | 10.43 |
| Pseudo-second order | $q_e^{calc}$ (mg g$^{-1}$) | 3.06 | 3.25 | 3.22 | 2.91 | 3.57 | 3.60 |
| | $k_2 \cdot 10^2$ (g mg$^{-1}$ min$^{-1}$) | 1.14 | 3.13 | 8.70 | 1.98 | 2.81 | 5.53 |
| | $R^2$ | 0.9950 | 0.9989 | 0.9996 | 0.9970 | 0.9981 | 0.9994 |
| | SE | 0.12 | 0.12 | 0.16 | 0.11 | 0.18 | 0.18 |
| | $\chi^2$ | 0.07 | 0.10 | 0.12 | 0.05 | 0.15 | 0.13 |
| Elovich | $\alpha$ (mg g$^{-1}$ min$^{-1}$) | 0.21 | 1.02 | 2.35 | 0.31 | 0.56 | 1.48 |
| | $\beta$ (g mg$^{-1}$) | 1.31 | 1.67 | 1.73 | 1.45 | 1.12 | 1.39 |
| | $R^2$ | 0.9773 | 0.9905 | 0.9036 | 0.9936 | 0.9633 | 0.9232 |
| | SE | 0.10 | 0.06 | 0.20 | 0.08 | 0.21 | 0.23 |
| | $\chi^2$ | 0.15 | 0.01 | 0.14 | 0.04 | 0.13 | 0.16 |
| Weber and Morris | $k_{id}$ (mg g$^{-1}$ min$^{-1/2}$) | 0.30 | 0.23 | 0.21 | 0.27 | 0.33 | 0.26 |
| | $I$ (mg g$^{-1}$) | 0.20 | 0.99 | 1.53 | 0.19 | 0.61 | 1.40 |
| | $R^2$ | 0.9950 | 0.9822 | 0.7477 | 0.9814 | 0.8740 | 0.7935 |
| | SE | 0.40 | 0.07 | 0.24 | 0.31 | 0.26 | 0.27 |
| | $\chi^2$ | 0.83 | 0.02 | 0.24 | 0.58 | 0.29 | 0.28 |

As it is listed in Table 3, in the case of the pseudo-second order kinetic model, the calculated values of $q_e$ fitted well with the experimental data, the $R^2$ values obtained for the pseudo-second order kinetics were closer to unity, and the conformity between the experimental data of calculated $q$ values was expressed by the least error values. The pseudo-second order kinetic model is more likely to predict the behavior of the whole experimental range of methylene blue and malachite green adsorption more than the pseudo-first order model. Thus, the experimental results support the assumption based on the pseudo-second order kinetic model that the rate-limiting step in the adsorption of the cationic dyes is chemisorption between the adsorbent and ions of the dyes [20].

Many studies on the kinetics of cationic dye adsorption on various citric acid-modified lignocellulosic materials have also shown higher correlations for the pseudo-second order model. The pseudo-second order equation generated the best agreement with experimental data for adsorption of methylene blue on citric acid-modified rice straw [11,12], citric acid-modified wheat straw [13,14], and citric acid-modified sesame straw [8] and for malachite green on citric acid-modified wheat bran [6] and citric acid-modified corn stalks [22].

The Elovich kinetic model is used to describe the ion exchange process where the activation energy has a great change [29]. In this study, the Elovich model for adsorption of the cationic dyes on CS-C generates a satisfactory fit to the experimental data. The values of $R^2$ of the Elovich equation are closer to unity, and conformity between the experimental data of calculated $q$ values expressed small error values (Table 3).

As the pseudo-second order and Elovich equations are successfully used to describe the adsorption kinetics of the cationic dyes on CS-C, it can be concluded that the adsorption is a chemical process with an ion exchange.

As can be seen from Table 3 in the case of the Weber and Morris model, the linear lines $q = f(t^{1/2})$ did not pass through the origin, and this deviation from the origin might be due to the difference in the mass transfer rate in the initial and final stages of adsorption [11]. It can therefore be concluded that the adsorption of the cationic dyes on CS-C is a complex

process involving both film diffusion and intraparticle diffusion inside the pores of the adsorbent. Moreover, it is found that correlation coefficient values in the Weber and Morris model at 333 K are low, showing inferior quality of linearization.

### 3.3. Equilibrium Studies

Adsorption isotherms of methylene blue and malachite green on CS-C at different temperatures were shown in Figure 3. The adsorption capacity of the cationic dyes on CS-C increased as the temperature increased from 293 to 333 K, indicating that the adsorption was an endothermic process. The maximum Langmuir adsorption capacity for methylene blue increased from 23.15 mg/g at 293 K to 26.60 mg/g at 333 K, and for malachite green, it increased from 21.41 mg/g at 293 K to 27.55 mg/g at 333 K. The endothermic nature of methylene blue and crystal violet adsorption has been observed on citric acid-modified wheat straw [13,14].

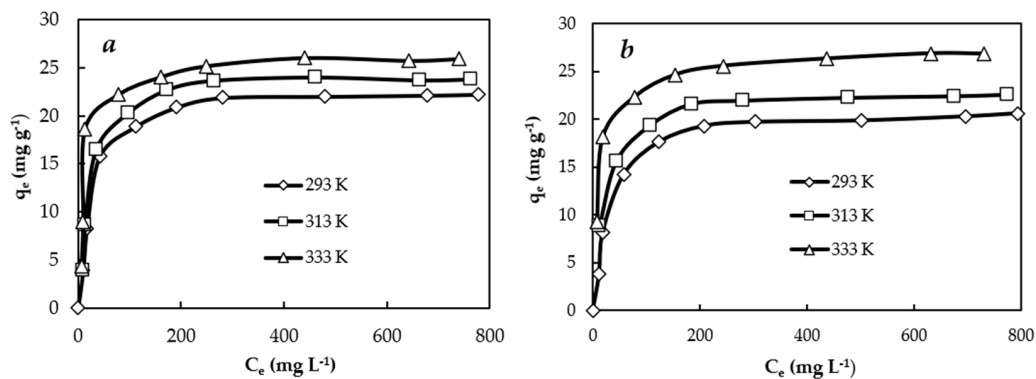

**Figure 3.** Adsorption isotherms of MB (**a**) and MG (**b**) on CS-C at different temperatures.

Adsorption isotherm models help to describe the interaction between an adsorbent and the given adsorbate, which is a critical factor in optimizing the use of adsorbents [17]. The calculated parameters of the Langmuir, Freundlich, and Temkin isotherms of the cationic dyes on CS-C are shown in Table 4.

The Langmuir model was more fitting compared to the other two models with regard to the higher values of $R^2$ and lower values of SE and $\chi^2$, indicating a possible monolayer adsorption of the cationic dyes on a surface of the adsorbent containing a finite number of identical sites under the concentration range studied [15,17]. Monolayer adsorption may involve chemical and physical adsorption [30].

The results indicate that the $R^2$ values were 0.6946–0.7878 for methylene blue and 0.7386–0.8020 for malachite green for the linear equation of the Freundlich isotherm (Table 4). Thus, these coefficient correlation values show bad linearity for the Freundlich isotherms of the cationic dyes on CS-C.

The Temkin model assumes that the heat of adsorption of all the molecules in the layer linearly decreases with coverage because of adsorbent–adsorbate interactions. As it is listed in Table 4, the correlation coefficients $R^2$ were 0.8226–0.8946 for methylene blue and 0.8759–0.9102 for malachite green for the Temkin isotherm linear equation. Nevertheless, these coefficient correlation values show good linearity. The good linear fitting of the Temkin isotherm to experimental data is an indication of the strong interaction between the cationic dyes and CS-C. The values of constants $K_T$ and $b$ increase with the increase in temperature.

**Table 4.** Comparison of the isotherm models.

| Isotherm Model | Parameter | MB | | | MG | | |
|---|---|---|---|---|---|---|---|
| | | **293 (K)** | **313 (K)** | **333 (K)** | **293 (K)** | **313 (K)** | **333 (K)** |
| Langmuir | $q_m$ (mg g$^{-1}$) | 23.15 | 24.81 | 26.60 | 21.41 | 23.53 | 27.55 |
| | $K_L \cdot 10^2$ (L mg$^{-1}$) | 3.6 | 4.0 | 5.8 | 3.0 | 3.6 | 5.4 |
| | R$^2$ | 0.9999 | 0.9982 | 0.9993 | 0.9989 | 0.9978 | 0.9996 |
| | SE | 1.16 | 1.52 | 2.86 | 0.88 | 1.58 | 1.93 |
| | $\chi^2$ | 1.08 | 1.76 | 5.33 | 0.73 | 2.15 | 2.37 |
| Freundlich | $1/n$ | 0.33 | 0.33 | 0.28 | 0.34 | 0.32 | 0.29 |
| | $K_F$ (mg$^{1-1/n}$ L$^{1/n}$ g$^{-1}$) | 3.05 | 3.36 | 4.95 | 2.69 | 3.25 | 4.80 |
| | R$^2$ | 0.7878 | 0.7560 | 0.6946 | 0.8020 | 0.7386 | 0.7545 |
| | SE | 4.11 | 4.75 | 5.13 | 3.74 | 4.17 | 4.84 |
| | $\chi^2$ | 7.76 | 9.64 | 12.99 | 6.02 | 8.40 | 10.29 |
| Temkin | $K_T$ (L g$^{-1}$) | 0.63 | 0.65 | 1.77 | 0.48 | 0.66 | 1.44 |
| | $b$ (kJ mol$^{-1}$) | 0.503 | 0.610 | 0.704 | 0.646 | 0.654 | 0.661 |
| | R$^2$ | 0.8946 | 0.8770 | 0.8226 | 0.9102 | 0.8759 | 0.8890 |
| | SE | 5.35 | 2.79 | 3.62 | 1.95 | 2.59 | 2.96 |
| | $\chi^2$ | 9.47 | 4.09 | 7.26 | 2.26 | 3.87 | 4.66 |

Comparing the values of R$^2$ and errors (SE, $\chi^2$) showed that linear fits using the Langmuir and Temkin isotherm equations were good for studying the adsorption of the cationic dyes on CS-C within the used concentration range, but the fit with the ones of the Langmuir isotherm equation was better.

*3.4. Thermodynamic Studies*

The thermodynamic parameters of adsorption ($\Delta G^o$, $\Delta H^o$, $\Delta S^o$) of methylene blue and malachite green on CS-C are listed in Table 5.

**Table 5.** Thermodynamic parameters of adsorption of the cationic dyes on CS-C.

| Dye | $T$ (K) | $K^o$ $10^{-4}$ | $-\Delta G^o$ (kJ mol$^{-1}$) | $\Delta H^o$ (kJ mol$^{-1}$) | $\Delta S^o$ (J K$^{-1}$ mol$^{-1}$) |
|---|---|---|---|---|---|
| MB | 293 | 6.1 | 26.8 | | |
| | 313 | 6.7 | 28.9 | 9.5 | 124 |
| | 333 | 9.8 | 31.8 | | |
| MG | 293 | 4.9 | 26.3 | | |
| | 313 | 5.9 | 28.6 | 11.8 | 130 |
| | 333 | 8.8 | 31.5 | | |

At all temperatures tested, the $\Delta G^o$ values were found to be negative, ensuring the spontaneous nature and feasibility of adsorption of methylene blue and malachite green on CS-C. The decrease in the value of $\Delta G^o$ with an increase in the temperature of the solution has demonstrated that adsorption of cationic dyes is favored at higher temperatures. The values of $\Delta H^o$ are positive, indicating that the adsorption reaction is endothermic. Based on the fact that dye effluents are usually produced at relatively high temperatures [8], the endothermic nature of the cationic dye adsorption on CS-C is advantageous in practice. The positive values of $\Delta S^o$ reflect an increase in the randomness at the solid/solution interface during the adsorption process.

*3.5. Desorption Studies*

An adsorbent with a fine regeneration capacity can reduce pretreatment costs and improve its reusability, which is of critical significance in practical applications for dye removal from wastewaters [19]. Desorption studies are also important to understand adsorption nature.

Strong binding forces such as covalent bonding result in chemical adsorption, and weak binding forces such as van der Waals forces result in physical adsorption. It is known that if the cationic dyes adsorbed onto the adsorbent surface can be desorbed by distilled water, the adsorption is dominated by weak bonds. If a solution of inorganic strong acid or base can desorb the dyes, then adsorption takes place as ion exchange. In this case, if a solution of weak organic acid can desorb the dyes, then the dyes are held by the adsorbent through chemisorption [11].

Solutions of a strong acid (HCl), a weak acid ($CH_3COOH$), a base (NaOH), and distilled water were chosen as eluents in a desorption study. The desorption studies of methylene blue and malachite green from CS-C by different eluents are presented in Figure 4.

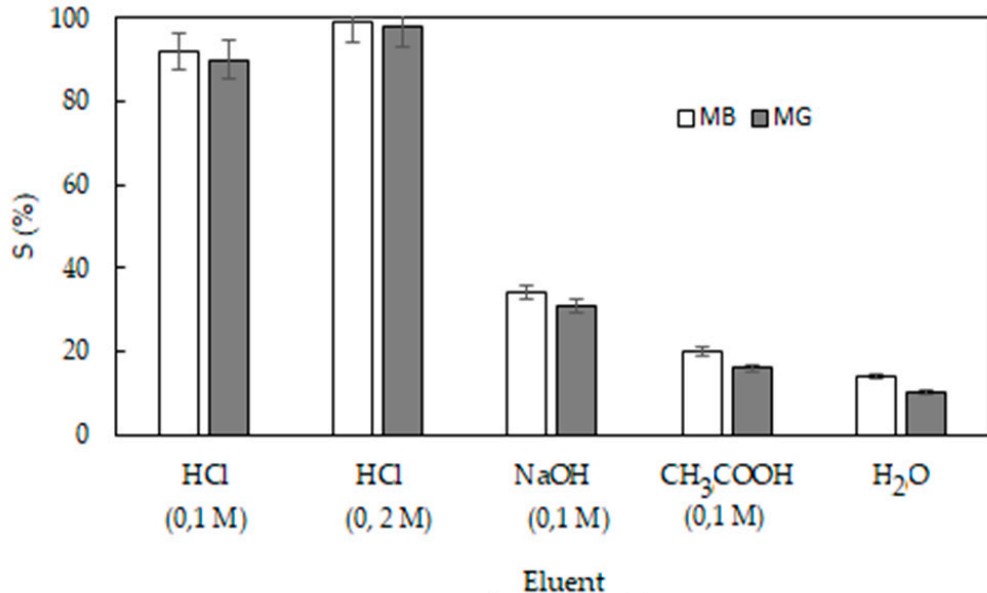

**Figure 4.** Desorption of the cationic dyes from dye-loaded CS-C by different eluents.

As seen from Figure 4, HCl solutions were the most efficient eluents to desorb the cationic dyes from the dye-loaded CS-C (Figure 4). It was observed that desorption percentages for methylene blue and malachite green by the HCl solution (0.1 M) were 92% and 90% and by the HCl solution (0.2 M) were 99% and 98%, respectively. The NaOH solution can withdraw 34% methylene blue and 31% malachite green, respectively. Desorption with distilled water and a solution of $CH_3COOH$ was obtained as follows: 14 % and 20% of methylene blue and 10% and 16% of malachite green, respectively. In the case of desorption with distilled water, the van der Waals forces between the phenolic ring of CS-C and the aromatic heterocyclic rings of the cationic dyes may also have contributed to the adsorption process [31]. Based on these findings, the desorption of the cationic dyes indicated that the dyes are adsorbed on CS-C by strong and weak forces.

In this study, the results showed that the process of methylene blue and malachite green adsorption on CS-C may be of a complex nature, consisting of both surface adsorption and intraparticle diffusion.

According to the aforementioned FT-IR analysis, citric acid was successfully incorporated in corn stalks. The surface of CS-C was apt to be charged positively or negatively at different pH solutions. As shown in Table 1, the $pH_{pzc}$ of CS-C is 3.5, indicating that the adsorbent was charged positively at pH < 3.5 and was charged negatively at pH > 3.5. The dye cations are positive in an aqueous solution. Therefore, the adsorption mechanism of cationic dyes on CS-C may include electrostatic interactions between surface carboxyl groups of the adsorbent and cationic dye ions. Furthermore, our kinetic and desorption

studies showed that ion exchange and chemisorption also promote the adsorption of methylene blue and malachite green on CS-C.

## 4. Conclusions

The present study focuses on the adsorption of methylene blue and malachite green from an aqueous solution using citric acid-modified corn stalks. The modification of corn stalks by citric acid due to the introduced free carboxyl groups increases the net negative charge on the corn straw fiber, thereby increasing its binding potential for cationic dyes. Equilibrium, kinetic, and thermodynamic studies were carried out for the adsorption of methylene blue and malachite green on citric acid-modified corn stalks.

Among the kinetic models tested (pseudo-first order, pseudo-second order, Elovich, and Weber–Morris models), the adsorption kinetics was best described by the pseudo-second order equation for the adsorption of methylene blue and malachite green on citric acid-modified corn stalks. The results of the Weber–Morris model suggested that intraparticle diffusion was not the only rate-controlling step of cationic dye adsorption on CS-C.

The equilibrium data have been analyzed using Langmuir, Freundlich, and Temkin isotherm models, and adsorption parameters for each isotherm were determined. It was shown that the Langmuir model was more suitable for methylene blue and malachite green adsorption on CS-C in the concentration range studied. It was determined that adsorption was positively affected by higher temperatures.

Thermodynamic parameters (such as changes in the standard Gibbs free energy, enthalpy, and entropy) were calculated, showing adsorption of methylene blue and malachite green adsorption on CS-C to be a feasible, spontaneous, and endothermic adsorption process. Hence, corn stalks, as a kind of discarded agriculture waste in many countries, could be used as an adsorbent after their citric acid modification according to the principle of so-called waste control.

**Author Contributions:** L.S. conceived the work and wrote the manuscript; M.Y. designed the experiments and performed the experiments; L.S. and M.Y. analyzed and interpreted the data; L.S. contributed to critical revision of the manuscript. All authors have read and agreed to the published version of the manuscript.

**Funding:** The study was carried out with the support of the Ministry of Education and Science of Ukraine.

**Conflicts of Interest:** The authors declare no conflict of interest.

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
