# Peer review of "Equilibrium, Kinetic, and Thermodynamic Studies of Cationic Dyes Adsorption on Corn Stalks Modified by Citric Acid"

_colloids, doi:10.3390/colloids5040052_

Round 1

Reviewer 1 Report

The paper reported on the Equilibrium, kinetic and thermodynamic studies of cationic dyes adsorption on corn stalks modified by citric acid. My comments for the manuscript as below;

In the introduction, please include why you need to study two cationic dyes in the study?

Have you calculated the degree of substitution of CS after modified with citric acid?

Punctuations need to be carefully checked. Some of the sentences missing comma and full stop.

Table caption for Table 3 should be above the table.

Have you studied the reproducibility of the CS-C as adsorbent for MB and MG?

Why desorption of MB and MG have very different percentage when using acid compared to alkaline solution?

Please include the error bar in Figure 4.

Author Response

Dear Reviewer,

Reviewer 2 Report

I have read the work of Soldatkina and Yanar entitled “Equilibrium, kinetic and thermodynamic studies of cationic 2 dyes adsorption on corn stalks modified by citric acid” submitted to Colloids and Interfaces. In my opinion, this work needs major revisions. Not all required experimental details are given, the paper lacks of originality. In addition, I cannot follow properly if authors used a background electrolyte or not, if the phase separation was done properly, if the calibration of the pH electrode was done at different temperatures, etc. My comments are listed below.

In general, I noticed too many issues with wording and phrasing in English (e.g. lines 11, 41, 42, etc.), please improve it.

Lines 26-27: If available, give some examples of the released concentrations of these dyes in the wastewaters.

In Table 1, it would be useful to provide the readers with the maximum adsorption capacity in the absence of citric acid as well.

Table 2: Instead of writing S, write Specific surface area. So the samples used in your current paper were exactly the same (same samples during one synthesis) as those used in the reference [24]? Or they were synthesized in the same way but correspond to different charges? If so, please re-characterize them (i.e. BET, pHPZC)

Section 2.2.2: with which detector was the IR instrument equipped? Over how many scans were the reported spectra averaged? What was the acquisition time? What was the spectral resolution? Which software did you use for data acquisition and evaluation?

Section 2.2.3: Were the batch experiments conducted under atmospheric conditions? If so, please write it explicitly. Which pH electrode have you used and which pH buffers did you take to calibrate your pH electrode? Did you use a background electrolyte to keep the ionic strength of your suspensions constant?

Line 133: Add the environmental relevance of performing all your batch studies at pH 6.

Equation 1 line 140: q is usually expressed in mg/g, as it is shown in your figure 2. According to your formula, the q value you calculated is dimensionless, which is sense less.

Lines 141-142: add the units of the different parameters.

Lines 143-145: How were the solid and liquid phases separated? By centrifugation? By filtration? Did you check the absence of colloids in the filtrates or supernatants, this can falsify your adsorption results. How was the calibration curve for the UV-vis measurements obtained, what was the concentrations of the different calibration samples?

Lines 157-159: add the units of the different parameters.

Equation 2 line 156: this should be a dimensionless parameter, it is apparently not the case.

Lines 170-174: add the units of the different parameters.

Lines 182-186: add the units of the different parameters.

Section 2.2.7: It is unclear to me how you calibrated your pH electrode at 313 and 333K, and how did you perform the solid/liquid phase separation at these elevated temperatures? Same question about the UV-vis measurements at 313 and 333K.

Section 2.2.7: please indicate that you assume the enthalpy and entropy of reaction to be constant from 293 to 333K.

Lines 194-197: add the units of the different parameters. Add the value of R.

Line 196: how are these s values calculated?

Lines 204-206: add the units of the different parameters.

Lines 209-211: Write FT-IR for all occurrences.

Lines 349-355: keep in mind that adsorption is a reversible process, both chemisorption and physisorption will exhibit a degree of reversibility, but to a different extent.

Section 3.6: don´t use such a title for this section. All the models you used are semi-empiric ones but to get really insight on the adsorption mechanism at a molecular level, you need to apply advanced spectroscopic techniques to characterize in situ the species formed at the solid/liquid interface and their stoichiometry, which is not what you have done in your study.

Line 382: write 3.5 and not 3,5.

Author Response

Dear Reviewer,

Round 2

Reviewer 2 Report

here are my comments:

Response 5:  Please also add the number of scans and the average time in the manuscript.
Lines 150-151: Please replace ‚containing‘ with ‚contacting‘.
Response 6: Please give the pH and supplier of the buffers.
Response 7: pH=6 is not an alkaline pH but rather circumneutral.
Response 9:  Equation 1: add unit of q in the text.
Response 10: Write 4,000 instead of 4000. You wrote that calibration curves were performed in the range of 1-10 mg/L. But above you wrote that you added 10 cm3 of dye solutions with different initial concentrations (20-1000 mg/L). Are you sure that all dye concentrations after sorption were ranging from 1-10 mg/L ?
Response 16: Section 2.2.7: please indicate that you assume the enthalpy and entropy of reaction to be constant from 293 to 333K. This was not done. 
